# DeTiME: Diffusion-Enhanced Topic Modeling using Encoder-decoder based LLM

**Weijie Xu, Wenxiang Hu, Fanyou Wu, Srinivasan H. Sengamedu**
**Amazon**
weijiexu@amazon.com

## Abstract

In the burgeoning field of natural language processing, Neural Topic Models (NTMs) and Large Language Models (LLMs) have emerged as areas of significant research interest. Despite this, NTMs primarily utilize contextual embeddings from LLMs, which are not optimal for clustering or capable for topic generation. Our study addresses this gap by introducing a novel framework named Diffusion-Enhanced Topic Modeling using Encoder-Decoder-based LLMs (DeTiME). DeTiME leverages Encoder-Decoder-based LLMs to produce highly clusterable embeddings that could generate topics that exhibit both superior clusterability and enhanced semantic coherence compared to existing methods. Additionally, by exploiting the power of diffusion, our framework also provides the capability to generate content relevant to the identified topics. This dual functionality allows users to efficiently produce highly clustered topics and related content simultaneously. DeTiME's potential extends to generating clustered embeddings as well. Notably, our proposed framework proves to be efficient to train and exhibits high adaptability, demonstrating its potential for a wide array of applications.

## 1 Introduction

Topic modeling methods, such as (Blei et al., 2003), are unsupervised approaches for discovering latent structures in documents and achieving great performance (Blei et al., 2009). These methods take a list of documents as input, generate a defined number of topics, and can further produce keywords and related documents for each topic. In recent years, topic modeling methods have been widely used in various fields such as finance (Aziz et al., 2019), healthcare (Bhattacharya et al., 2017), education (Zhao et al., 2021a,b), marketing (Reisenbichler, 2019), and social science (Roberts et al., 2013). With the development of Variational Autoencoder (VAE) (Kingma and Welling, 2013),

the Neural Topic Model (Miao et al., 2018; Dieng et al., 2020) has attracted attention due to its better flexibility and scalability. The topic is generated through the reconstruction of the bag-of-word representations of the document (Miao et al., 2018).

The progress of large language model (LLM) (Vaswani et al., 2017; Radford et al., 2019) brings significant advancements in the NLP community. Sentence embedding is the process of converting sentences into numerical vectors in a high-dimensional space. LLM-based sentence embedding has been applied to topic modeling by using it to reconstruct bag of word representation of documents (Bianchi et al., 2021a), to cluster document directly (Grootendorst, 2022) or both (Han et al., 2023). Sentence embedding-based models have been shown to achieve high performance regarding coherence and diversity (Zhang et al., 2022). Embeddings with higher clusterability are likely to perform well in classification tasks. However, *sentence embeddings are in general not perform well in clustering*. The best performed sentence embedding has an average v-measure (Rosenberg and Hirschberg, 2007) below 0.44 even if it uses kmeans and set the cluster equal to the number of different labels (Muennighoff et al., 2022). This means that their clusterability can be even lower when the latent dimension increases. Lastly, language modeling is a powerful generative tool (Brown et al., 2020). *While topic modeling has been utilized for generation (Wang et al., 2019), its integration with Large Language Models (LLMs) for generation remains less explored.*

In this study, we introduce DeTiME, an innovative topic modeling framework that exploits the capabilities of the encoder-decoder Large Language Model (LLM). Specifically, we design a task to train an adapted encoder-decoder LLM, as depicted in Figure 2. We generate an embedding using this architecture, which exhibits high clusterability

compared to established models as illustrated in Figure 1. Furthermore, we design a topic modeling approach using the last hidden layer of our modified LLM encoder as input. This technique notably outperforms standard methods across all pertinent metrics. Additionally, we leverage diffusion and our proposed framework to generate relevant documents. Our major contributions are as follows:

1. We modify the encoder-decoder LLM and design a task to create an embedding ideal for topic modeling, even using a smaller model.

2. The fabricated embeddings outperform existing methods in terms of clusterability

3. We devise a topic modeling method based on the embedding that achieves superior results in both clusterability and semantic coherence, compared to the relevant topic modeling methods.

4. We demonstrate the ability to produce relevant content based on this model by harnessing diffusion, indicating potential practical applications.

5. Our framework exhibits flexibility as it can be seamlessly adapted to various encoder-decoder LLMs and neural topic modeling methods, broadening its applicability in the field.

By documenting detailed methodology and empirical results, we aim to inspire further research in this domain, and provide a strong foundation for future work on topic modeling and LLMs.

## 2 Related work

### 2.1 Language Modeling

Recent transformer-based models, such as BERT (Devlin et al., 2019), GPT-3 (Brown et al., 2020), and GPT-4 (OpenAI, 2023) have achieved unmatched performance in numerous language tasks. Utilizing self-attention mechanisms, they capture context from both past and future tokens, generating coherent text. These rapidly evolving Large Language Models (LLMs) carry significant implications for diverse sectors and society. T5 (Raffel et al., 2020) treats every NLP task as a text-to-text problem, using a standard format with input and output as text sequences. It employs an encoder-decoder framework and is pretrained

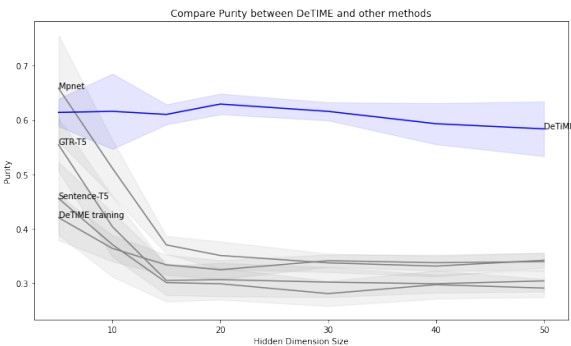

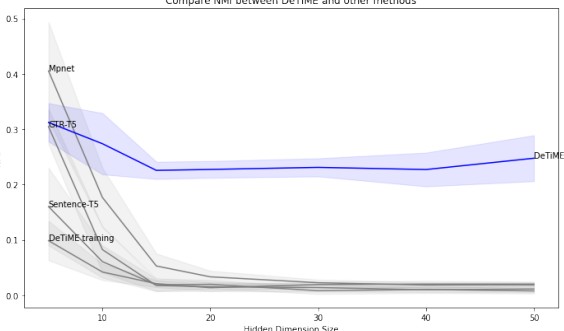

Figure 1: A summary of a few of our findings: (1) Our embeddings outperform the best clusterable methods (selected from (Muennighoff et al., 2022)). (2) The same framework with a slightly different finetuned task(DeTiME Training) does not perform well. (3) When compressed, our embeddings excel in higher dimensions, making them ideal for topic modeling. Detailed settings is in Appendix E.

on extensive datasets. FlanT5 (Chung et al., 2022) enhances T5 by finetuning instructions across multiple datasets. Compared to encoder only (Bert) or decoder only model(GPT), encoder-decoder models such as FlanT5 allow the encoder to extract vital input information for output generation (Rothe et al., 2020).

Prefix tuning (Li and Liang, 2021) modifies a fixed-length "prefix" of parameters prepended to the input during fine-tuning, significantly reducing the number of parameters required. This efficiency doesn't compromise performance; it often matches or surpasses traditional fine-tuning methods across various NLP tasks. The technique enables the model to learn task-specific initial hidden states for LLM, steering the generation process appropriately without hindering the model's generality due to the fine-tuning task.

### 2.2 Sentence Embedding

Contextual embeddings aim to encode sentence semantics in a machine-readable format. Word embeddings like Word2Vec (Mikolov et al., 2013)

and GloVe (Pennington et al., 2014) capture word-level meaning but struggle with larger text structures. Advanced models like the Universal Sentence Encoder (USE) (Cer et al., 2018) and InferSent (Conneau et al., 2018) were developed to better capture sentence nuances. USE employs transformer or Deep Averages Networks, while InferSent uses a bidirectional LSTM with max pooling. Sentence-BERT (Reimers and Gurevych, 2019) utilizes siamese BERT-Networks. However, *these models often struggle to capture context-dependent sentence meanings, resulting in lower clusterability*. This might be due to their reliance on contrastive loss on sentence pairs, which might focus on specific similarities rather than the overall semantic relationship.

## 2.3 Topic Modeling

The Neural Topic Model (NTM) (Miao et al., 2016) employs variational inference but struggles with semantics and interpretability, while the Embedding Topic Model (ETM) (Dieng et al., 2019) uses pre-trained word embeddings to capture semantics. However, *NTMs rely on bag-of-word representations, limiting their ability to capture document semantics effectively*.

The Contextual Topic Model (CTM) (Bianchi et al., 2021a) uses sentence embeddings and bag of words as input to reconstruct bag of words embeddings, while BERTopic (Grootendorst, 2022) combines sentence embedding and clustering techniques like UMAP and HDBSCAN for topic generation. Other models (Han et al., 2023) use both clustering techniques and reconstruction to create high-quality topics. Nonetheless, *contextual embedding based topic modeling methods lack a reconstruction process or only reconstruct bag of words representations*. These disadvantages limit its ability to generate relevant content. We examined other related works in Appendix H

## 2.4 Diffusion

Drawing inspiration from non-equilibrium thermodynamics, the diffusion model adds noise to the data distribution in a forward process and learns a reverse denoising process (Sohl-Dickstein et al., 2015). (Song and Ermon, 2020) further applied this for high-quality image generation, comparable to leading likelihood-based models and GANs (Goodfellow et al., 2014), but with more stable training and generation due to iterative diffusion.

Denoising Diffusion Probabilistic Models (DDPM) (Ho et al., 2020) have garnered attention for generating high-quality samples sans adversarial training, sometimes surpassing other generative models. Speedier sampling was achieved in (Song et al., 2022) with denoising diffusion implicit models. The success of image generation models like CLIP (Radford et al., 2021), Stable Diffusion (Rombach et al., 2022), and Midjourney (Oppenlaender, 2022) leveraged such diffusion-based methods. Their use extends to NLP tasks including natural language generation, sentiment analysis, and machine translation (Zou et al., 2023). It has also demonstrated that the diffusion model is able to generate high-quality text from noise samples in the continuous embedding space(Li et al., 2022; Gong et al., 2023; Gao et al., 2022; Lin et al., 2023b). Yet, *diffusion hasn't been used for topic modeling as a content generation tool*.

## 3 Methods

The goal of this paper is to create a framework that leverages encoder-decoder LLM to generate topics that is highly clusterable and able to generate topic related sentence. To achieve that, we need to create an embedding that could be used to generate text as well as be highly clusterable. Thus, we designed a specific task and dataset for our use case. We add CNN encoder and decoder on top of FlanT5 to generate that can easily fit into neural topic modeling for further dimension reduction. We further design a variational autoencoder to take the output of the CNN encoder as input and generate topics and reconstruct embeddings. This is achieved by 2 autoencoders. The first autoencoder is a variational autoencoder which generates topic distribution and reconstructs bag of words representations. To reconstruct the embeddings from $enc_2$. We use another autoencoder to generate embeddings from topic distribution and reconstructed bag of words. The detailed structure and name are in Figure 3. We do not train or finetune FlanT5 and CNN during the topic modeling process which makes our methods cost-effective. We then leverage diffusion to generate high quality text that represents the document.

This section contains four components. First, we present the dataset and define the finetuned task. Second, we elaborate on our modified FlanT5 and the fine-tuning strategy. The third component introduces a variational autoencoder designed for topic

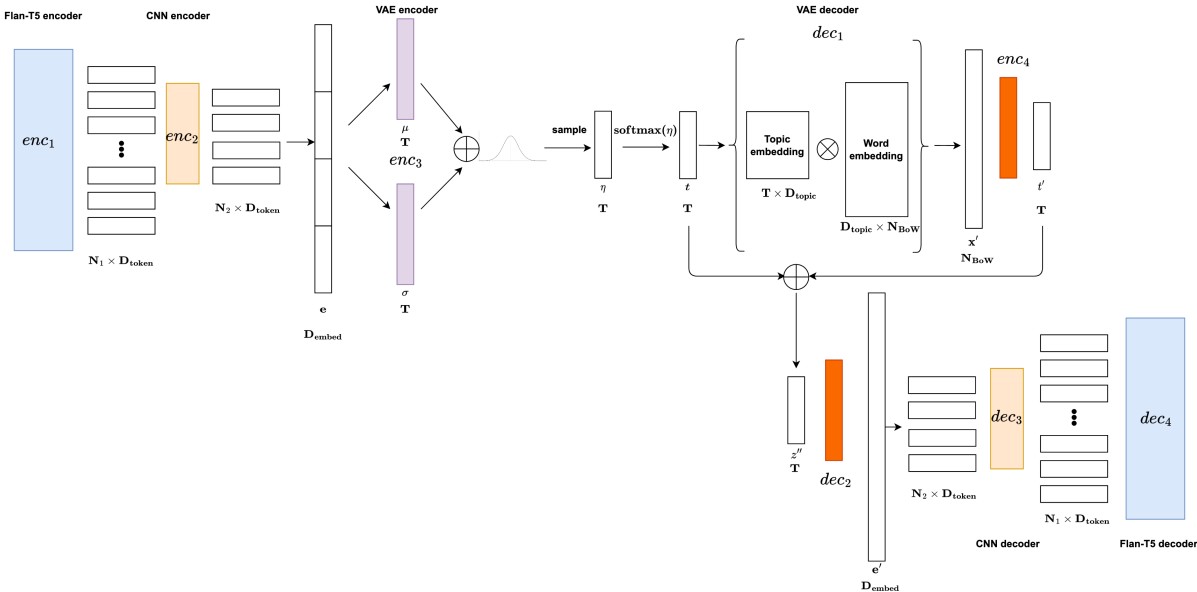

Figure 2: DeTiME framework. We have 4 encoders and 4 decoders. $enc_1$ and $enc_2$ are compressing the input document to the lower dimension. $enc_3$ is to construct topic distribution. $dec_1$ is to reconstruct bag of words representations. $enc_4$ is to extract the hidden dimension from the reconstructed bag of words. $dec_2$, $dec_3$ and $dec_4$ is to reconstruct/rephrase the input document. In our method, we name the number of dimensions for embedding $D_{token}$ and maximum sequence length $N_1$. The dimension of the compressed vector is $D_{embed}$. The number of topics equals $T$. The dimension of vocabulary is $N_{BoW}$. The dimension of topic embeddings is $D_{topic}$.

modeling and generation. Finally, we utilize diffusion to generate content relevant to the derived topics.

### 3.1 Tasks and Finetune Dataset

To achieve effective topic modeling methods, we aim to generate embeddings that are highly clusterable and capable of generating document-relevant topics. We utilize a paraphrase dataset in which the input and output sentences are equivalent in meaning. Such equivalent sentences should belong to similar topics, thereby aiding us in generating similar sentences. In contrast to methods that use the same sentence for both input and output, our task assists the language model in learning the semantic meaning of sentences rather than simply memorizing the embeddings. As illustrated in Figure. 1, the DeTiME-training model represents the model generated by the task where the input and output are identical contents. As you can see, the clusterability of this method is substantially lower than ours. Thus, **rephrase task is effective to generate clusterable contents.** Moreover, the paraphrase task is not sufficiently easy (Vahtola et al., 2022) and is less likely to impair the utility of the language model. We concatenate similar sentence pairs from the STS benchmark, a dataset

for comparing meaning representations, to form our dataset (Agirre et al., 2012, 2013, 2014, 2015, 2016). We select pairs with scores above 80 percent of the maximum, yielding a total of 22,278 pairs. This dataset addresses the limitations of existing paraphrase datasets, which are either domain-specific (Dolan and Brockett, 2005; Gohsen et al., 2023), or generated by potentially unreliable language models (Shumailov et al., 2023). Our composite dataset is diverse, including data from news, captions, and forums.

### 3.2 Modified Encoder Decoder LLM

The motivation for this nested autoencoder structure stems from the limitation of existing sentence embeddings, which struggle to reconstruct sentences as they are primarily trained using contrastive learning (Reimers and Gurevych, 2019) rather than reconstruction. In other words, similar sentences are distributed close to each other in the learned embedded vector space. We choose an encoder-decoder model due to its ability to preserve essential information through encoding process. Specifically, encoder-decoder approaches, like T5, encapsulate vital information in the encoder's final hidden state. We can compress this final hidden state to create our embeddings. FlanT5 (Chung

et al., 2022) outperforms T5 in standard tasks by leveraging a (Wei et al., 2023) and instruction fine-tuning (Chung et al., 2022). We believe that the final hidden layer of a fine-tuned FlanT5 can represent the input information.

The purpose of CNN is to compress output from FlanT5 encoder to create embeddings for topic modeling as illustrated in Append F. Using the encoder output as an embedding leads to excessive length and dimensionality, causing sparsely distributed vectors, poor clusterability, and issues in downstream tasks like topic modeling. To address this, we incorporate a variational autoencoder to reconstruct FlanT5's final encoder hidden layer. We trained MLP, RNN, and CNN-based autoencoders, but MLP introduced too many parameters and underperformed. LSTM, bidirectional LSTM, and GRU (Sherstinsky, 2020), with varied attention schemes (Xia et al., 2021), mostly yielded empty results or identical output embeddings, likely due to the FlanT5 encoder's non-sequential information processing. Applying a 1D convolution on the sequence dimension allowed for dimensionality reduction, with nearby embeddings showing high correlation, suggesting possible compression using a convolutional network on the sequence side. **We can adapt the same framework to other existing encoder decoder LLM** such as BART (Lewis et al., 2019).

We utilize Parameter Efficient Fine-tuning (PEFT) because it reduces the number of parameters to be fine-tuned, making the process more efficient and often yielding comparable or even superior performance to traditional fine-tuning (Liu et al., 2022). We adopt prefix fine-tuning (Li and Liang, 2021) in our work. During fine-tuning, we train both prefix fine-tuning related parameters and the CNN-based autoencoder for the paraphrase tasks. We then use the output from the CNN-based autoencoder's encoder for downstream topic modeling tasks. In our experiment, we use a relatively small model FlanT5 base (248M parameters) to illustrate the effectiveness of our framework.

### 3.3 VAE structure for topic modeling

Our VAE serves two purposes. First, it generates a highly clusterable topic distribution. Second, it reconstructs the output of the CNN encoder $e$, enabling it to be input into the decoder of the CNN autoencoder. Prior research (Srivastava and Sutton, 2017) demonstrated that a Variational Autoencoder

(VAE) aiming to reconstruct a bag of words produces high-quality topic embeddings. Our VAE has two encoders and two decoders. $enc_3$ is used to encode the output of the CNN encoder ($e$) into a topic distribution $t$. $enc_3$ has two parts: the first is a multi-layer perceptron (MLP) that maps the input to a lower dimension, and the second consists of two MLPs to generate the mean and the log of the standard deviation vector of size T: $\mu, log(\sigma) = enc_3(e)$. We sample a latent representation using the mean and standard deviation: $\eta \sim N(\mu, \sigma)$, and apply a softmax function to generate the topic distribution $t = softmax(\eta)$.

The $dec_3$ is used to decode the topic distribution $t$ into a bag-of-words representation $X'$. Existing research (Dieng et al., 2020) shows that topic-word similarity matrix offers better quality in reconstructions. The decoder consists of two matrices. We use a vocabulary embedding matrix $e_V \in R^{D_{Topic} \times N_{BoW}}$, where $D_{Topic}$ represents the dimension of word embeddings and $N_{BoW}$ represents the dimension of the vocabulary. The decoder $\phi$ learns a topic embedding matrix $e_T \in R^{T \times D_{Topic}}$. The topic-to-word distribution is denoted as

$$E = softmax(e_T e_V^T) \tag{1}$$

$$X' = t \times E \tag{2}$$

Here, $X'$ represents the reconstructed bag of words. The product of the generated topic distribution and this matrix $E$ yields a bag-of-words reconstruction.

The $enc_4$ is a neural network that encodes the generated bag of words back to a vector $t'$, having the same dimension as the topic embeddings dimension: $t' = enc_4(X)$. We add residual connections between two compressed vectors and use a neural network to generate input embeddings:

$$e' = dec_4(t + t') \tag{3}$$

It's necessary to reconstruct input embeddings ($e$) to be fed into the decoder to reconstruct the rephrased input sentence. We believe that the reconstructed bag of words can enhance the ability of sentence reconstruction. The residual connection helps the model leverage both the reconstructed bag of words and topic distribution to reconstruct input embeddings. This simplifies our input embedding reconstruction and ensures that the topic embeddings can capture semantic information from the output of the CNN decoder $e$. Our VAE leverages

only bag of words representations and contextual embeddings. **Our VAE can also take other contextual embeddings as input.** Our loss function has three components: the reconstruction loss for the bag of words, the reconstruction loss for input embeddings using mean square error, and the KL Divergence for the normal distribution. The loss for a single input $e$ is as follows:

$$L = -Xlog(X') + (e - e')^2 + KL(t|N(\mu, \sigma)) \tag{4}$$

### 3.4 Diffusion for content generation

Our pretrained model can compress the text and embed them in a low-dimensional space while keeping the semantic information and high-quality clustering. It is natural to wonder if this pretrained model can be used to generate topic-guided text. One of the challenges is that the decompression process in the pretrained model may induce noise, loss some information and thus the quality of the generated text will be impacted. Specifically, the latent dimension (i.e. the vector space of $z''$ before the $dec_2$ in Figure 3) is several orders of magnitude lower than the dimension of embedding vector $e'$ in DeTiME. When we reconstruct text from latent vectors, it may hugely deviate from any reasonable input for FlanT5 decoder $dec_3$.

To overcome this, we have leveraged the diffusion models to denoise the generated text embedding from the topic modeling with structure as shown in Figure 3. It has demonstrated that the diffusion model is able to generate high-quality text from noise samples in the continuous embedding space (Li et al., 2022; Gong et al., 2023; Gao et al., 2022; Lin et al., 2023b). In the training component, we employ a DDPM-scheduled Autoencoder with residual connections as the diffusor (Ho et al., 2020) in the text embedding continuous space (i.e. the space after $enc_2$ in Figure 3) using the embedded vectors obtained from the pretrained model. Specifically, during the forward process, the Gaussian noises is gradually added to $X_0$ according to a variance schedule $\beta_1, ..., \beta_T$, the noisy sample at time step $t$ is expressed as

$$q(X_t|X_0) = N\left(X_t; \sqrt{\bar{\alpha}_t}X_0, \sqrt{1-\bar{\alpha}_t}I\right) \tag{5}$$

where $\bar{\alpha}_t = \Pi_{i=1}^t \alpha_i$ with $\alpha_i = 1 - \beta_i$. Our diffusor is trained to minimize the squared error between the predicted and true noise. The predicted noise $z(X_t, t)$ at time step $t$ is obtained by the diffusor as following:

$$
\begin{aligned}
z^1 &= X_t + Sinusoid(t) \\
z^2 &= FC_1^{COMP}(z^1) \\
z^3 &= FC_2^{COMP}(z^2) \\
z^4 &= FC_3(z^3) \\
z^5 &= FC_4^{RECONST}(z^4 + z^3) \\
z(X_t, t) &= FC_5^{RECONST}(z^5 + z^2).
\end{aligned} \tag{6}
$$

This diffusor consists of 2 fully connected layers $FC^{COMP}$ to compress the input and 2 fully-connected layers $FC^{RECONST}$ to reconstruct. We also add residual connections between compress and reconstruct layers. Similar to UNet (Ronneberger et al., 2015), the Sinusoidal positional embeddings $Sinusoid(t)$ is used to encode time.

Then, in generating component, this trained diffusor is used to denoise the embedding after the $dec_2$ in Figure 3. The intuition behind this denoising process is as follows. The forward process of diffusion itself is a process that converts the unknown and complex data distribution into one (normal distribution in our case) that is easy to sample from. By adding back the learned noise with small iterative steps, we are able to take a sample from the noise subspace (support a simple distribution) to the data subspace (support the unknown data distribution). Similarly, for an embedding obtained from the topic modeling that deviates from the embedding distribution corresponding to the unknown input data distribution, we should also be able to take this embedding back to the area supporting the original embedding distribution.

## 4 Experimental Results

### 4.1 Topic Modeling

**Dataset** Our experiments are conducted on labeled benchmark datasets for topic modeling: **AgNews** (Zhang et al., 2016), **20Newsgroups** (Lang, 1995) and **bbc-news** (Greene and Cunningham, 2006). The average document length varies from 38 to 425. We use the text as it is for the contextual embedding generation. To get bag of words, we use the word tokenizer from nltk to tokenize, remove digits and words with lengths less than 3, and remove stop words and words that appear less than 10 time. Additional details on the dataset and places to download processed data are available in Appendix B.

| Methods | Purity | NMI | Km-Purity | Km-NMI | diversity | $C_v$ |
|---|---|---|---|---|---|---|
| ETM | **0.4677 ± 0.04** | 0.2502 ± 0.07 | 0.4063 ± 0.07 | 0.2400 ± 0.08 | 0.4177 ± 0.05 | 0.5594 ± 0.01 |
| GSM | 0.2701 ± 0.02 | 0.0687 ± 0.03 | 0.3167 ± 0.03 | 0.1312 ± 0.03 | 0.2991 ± 0.01 | 0.3495 ± 0.01 |
| vONT | 0.3727 ± 0.02 | 0.1604 ± 0.03 | 0.4941 ± 0.05 | 0.2688 ± 0.05 | 0.5937 ± 0.06 | 0.5151 ± 0.01 |
| NVDM | 0.4254 ± 0.04 | 0.2373 ± 0.07 | 0.3768 ± 0.07 | 0.2138 ± 0.05 | 0.2633 ± 0.05 | 0.4715 ± 0.02 |
| ZTM | 0.3637 ± 0.003 | 0.1019 ± 0.003 | 0.3479 ± 0.003 | 0.1087 ± 0.001 | 0.6796 ± 0.03 | 0.6705 ± 0.02 |
| CTM | 0.4307 ± 0.03 | 0.1641 ± 0.04 | 0.4191 ± 0.04 | 0.1819 ± 0.05 | **0.7198 ± 0.01** | 0.6966 ± 0.02 |
| DeTiME bow | 0.3416 ± 0.004 | 0.1300 ± 0.009 | 0.5007 ± 0.03 | 0.2591 ± 0.02 | 0.5362 ± 0.04 | 0.7186 ± 0.004 |
| DeTiME resi | 0.3239 ± 0.01 | 0.1098 ± 0.01 | 0.4230 ± 0.01 | 0.1741 ± 0.02 | 0.5802 ± 0.01 | **0.7435 ± 0.002** |
| DeTiME | 0.4577 ± 0.03 | **0.2983 ± 0.03** | **0.5929 ± 0.04** | **0.3463 ± 0.05** | 0.6913 ± 0.02 | 0.7203 ± 0.01 |

Table 1: The main results for all clusterability metrics, diversity, and coherence ($C_v$). The number of topics is 20. The best and second-best scores of each dataset are highlighted in boldface and with an underline, respectively. The result represents the average value obtained from three datasets, where each dataset was processed 10 times to compute the mean and standard deviation.

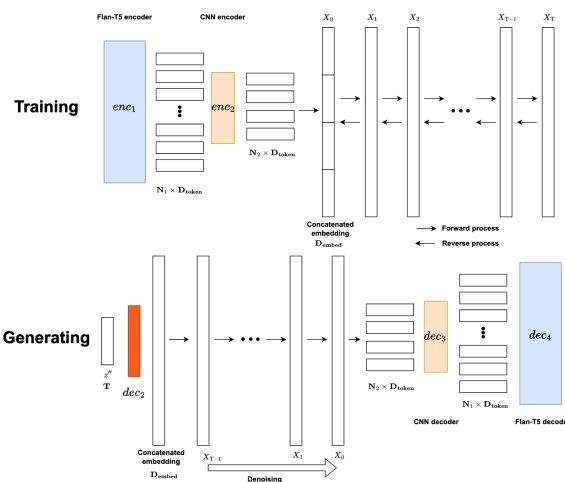

Figure 3: The diffusion framework based on the main framework in Figure 2. In the training component, a DDPM-scheduled Autoencoder with residual connections diffusor is trained using the embedding vectors obtained from the $enc_2$. In generating part, the trained diffusor is used to denoise the embedding vectors transformed from the topic vectors hidden space before the text generation. It's important to note that we normalized the hidden space before passing it to the $dec_2$.

**Baseline Methods** We compare with common NTM methods and contextual embedding based methods. We explain the reasons for choosing these methods in Appendix D. These methods include: **NVDM** (Wang and YANG, 2020), VAE architecture for topic modeling with the encoder is implemented by multilayer perceptron, the variational distribution is a Gaussian distribution; **GSM** (Miao et al., 2018), an NTM replaces the Dirichlet-Multinomial parameterization in LDA with Gaussian Softmax; **ETM** (Dieng et al., 2020), an NTM model which incorporates word embedding to model topics; **vONT** (Xu et al., 2023e), a vMF based NTM where they set the radius of vMF distribution equal to 10; **CTM** (Bianchi et al., 2021b) trains a variational autoencoder to reconstruct bag of words representation using both contextual embeddings as well as bag of words representation. **ZTM** (Bianchi et al., 2021b) is similar to CTM but only use contextual embeddings; **DeTiME resi** is the DeTiME model with out residual connections. The reconstruction of embedding is hugely dependent on the reconstructed bag of words; **DeTiME bow** is the DeTiME model without reconstruction of bag of words and $t'$ is used to represent topics.

**Settings** In our experiment setting, The hyperparameter setting used for all baseline models and DeTiME is the same as (Burkhardt and Kramer, 2019). For neural topic modeling and our encoder and decoder, we use a fully-connected neural network with two hidden layers of half of the hidden dimension and one quarter of hidden dimension and GELU (Hendrycks and Gimpel, 2023) as the activation function followed by a dropout layer. We use Adam (Kingma and Ba, 2017) as the optimizer with learning rate 0.001 and use batch size 256. We use (Smith and Topin, 2018) as scheduler and use learning rate 0.001. We use 0.0005 learning rate for the DeTiME bow because the loss may

overflow when the learning rate is 0.001. We use word embeddings (Mikolov et al., 2013) to represent word embeddings on the dataset for vONT, ETM, and DeTiME and keep it trainable for DeTiME. For vONT, we set the radius of the vMF distribution equal to 10. For CTM and ZTM, we use all-mpnet-base-v2 as our embeddings since it performs the best in clusterability in Figure 1. We use the same way to find key words as suggested by CTM. Our code is written in PyTorch and all the models are trained on AWS using ml.p2.8xlarge (NVIDIA K80). Detailed code implementations for methods and metrics are in Appendix C

**Evaluation Metrics** We measure the topic clusterability, diversity, and semantic coherence of the model. To measure clusterability, we assign every document the topic with the highest probability as the clustering label and compute **Top-Purity** and Normalized Mutual Information(**Top-NMI**) as metrics (Nguyen et al., 2018) to evaluate alignment. Both of them range from 0 to 1. A higher score reflects better clustering performance. We further apply the KMeans algorithm to topic proportions z and use the clustered documents to report purity(**Km-Purity**) and NMI **Km-NMI** (Zhao et al., 2020). We set the number of clusters to be the number of topics for the KMeans algorithm. Topic coherence($C_v$) uses the one-set segmentation to count word co-occurrences and the cosine similarity as the similarity measure. Compared to other metrics, $C_v$ is able to capture semantic coherence. We only benchmark $C_v$ because most of coherence metrics are similar to each other (Lim and Lauw, 2023). For **diversity**, we measure the uniqueness of the words across all topics divided by total keywords. For each topic, we set the number of keywords equal to 25. Furthermore, we run all these metrics 10 times. We report averaged mean and standard deviation. We also include evaluations on Perplexity in Appendix G

**Results** The experiment shows that DeTiME outperforms all other methods in NMI, Km-NMI, and Km-Purity, which underscores its ability to **generate highly clusterable topic distributions**. Furthermore, DeTiME has the second highest scores in coherence(The highest score is also a DeTiME variation), **validating the exceptional semantic coherence of topics generated from our methods**. Observations reveal that the CTM and DeTiME's high diversity scores highlight the benefit of incorporating bag of words inputs, enhancing diversity

performance. By eliminating the bag of words reconstruction components, we found a decrease in diversity and clusterability, indicating the importance of this component in boosting purity and NMI. When we removed the residual connection, we observed an improvement in coherence but a decrease in clusterability. This trade-off suggests that the absence of a residual connection may prevent the topic distribution from effectively capturing the information from embeddings, thus reducing clusterability. DeTiME resi performs better than ZTM in clusterability related metrics, which confirms that **our embedding is more clusterable than existing sentence embeddings**.

## 4.2 Diffusion for content generation

To evaluate how the diffusor improves the quality of the generated text, we compared the generated text before and after the diffusion. Specifically, we utilized the Flesch Reading Ease (FRE), Flesch-Kincaid Grade Level (FKGL), and Dale-Chall Readability Score (DCRS) to measure the readability of the generated text before and after the diffusion (Goldsack et al., 2022). In general, a higher FRE (lower FKGL and DCRS) indicates that the text is easier to read. In this experiment, we generated 1000 random topic vectors and passed them to $dec_2$, then the denoising process is followed to generate text. The main results are shown in Table 2. As observed, after the denoising process, the FRE increases significantly across all datasets, which indicates that diffusion makes the content easier to understand. Meanwhile, the value of FKGL and DCRS decreases from $T = 500$ to $T = 1000$. One of the reasons for the low score of FKGL and DCRS at $T = 0$ is that some of the samples contain only repeated words, making them easy to understand. Overall, after more steps in diffusion, the generated text becomes more readable for a lower grade. This experiment demonstrates that **our generated content achieves higher readability, indicating the potential of our framework to generate topic-relevant content.**

**Human Evaluation** To ensure the generated content is valuable to humans, a human evaluation was conducted with regard to the text generated after diffusion, as seen in Figure 3. In this evaluation, we generated 400 pieces of text. Each piece was evaluated for fluency, grammar, and redundancy by three different human annotators, as suggested by (Celikyilmaz et al., 2021). We com-

| Datasets | 20Newsgroups | | | bbc-news | | | AgNews | | |
|---|---|---|---|---|---|---|---|---|---|
| Time point | $T = 0$ | $T = 500$ | $T = 1000$ | $T = 0$ | $T = 500$ | $T = 1000$ | $T = 0$ | $T = 500$ | $T = 1000$ |
| FRE | -25.9600 | 51.1390 | 54.2467 | 6.8600 | 36.5889 | 60.9407 | 36.6200 | 64.1707 | 63.1074 |
| FKGL | 53.2000 | 10.7017 | 9.8955 | 30.2000 | 12.6860 | 9.1856 | 8.4000 | 9.0876 | 8.6781 |
| DCRS | 7.3500 | 8.4758 | 7.8822 | 4.0100 | 8.3304 | 8.2010 | 66.8500 | 8.1890 | 8.1059 |

Table 2: The average readability scores at different time steps during the denoising process. A general increase in readability is observed.

pared our results with a baseline through t-tests and found that the generated text exhibited fluency and grammatical correctness with statistical significance ($p < 1e - 14$). This demonstrates that **our generated contents are of high quality**. More details about the survey setup, results, and examples of generated text can be found in Appendix A.

## 5  Conclusion and Future Work

We have developed a framework DeTiME for generating highly clusterable embeddings, leveraging the strengths of paraphrase tasks, FlanT5, and CNN. In addition to this, we introduced a variational autoencoder structure capable of reconstructing embeddings while simultaneously producing highly coherent, diverse, and clusterable topics. Our design incorporates a diffusion process for generating content that provides representative depictions of various topics. The flexibility of our embedding generation structure allows for easy adaptation to other encoder-decoder language model architectures, eliminating the need for retraining the entire framework, thereby ensuring cost-effectiveness. Additionally, our variational autoencoder structure is versatile, and capable of being applied to any contextual embeddings. Other methods could further improve with larger LLM.

Moving forward, we aim to further improve the performance of our embeddings by training on larger models such as Flan-T5-XL. Benchmarking other Pre-training with Fine-Tuning (PEFT) methods, such as LORA, may also enhance our system's performance. Given the high clusterability of our embeddings, we plan to extend our work to semi-supervised document classification (Xu et al., 2023b,c; Balepur et al., 2023; Lin et al., 2023a). This framework could be applied to identify the most representative documents within extensive document collections. This functionality could make our model suitable for generation topic guided generation (Xu et al., 2023a) Finally, we envisage utilizing this framework to generate superior summarizations for large documents. This could be achieved by training a decoder for summarization, generating a summarization for each topic, and subsequently concatenating them. This framework can also be extended to hierarchical topic modeling (Chen et al., 2023; Shahid et al., 2023; Eshima and Mochihashi, 2023), mitigate data sparsity for short text topic modeling (Wu et al., 2022), generate topic-relevant and coherent long texts (Yang et al., 2022), and construct a network of topics together with meaningful relationships between them (Byrne et al., 2022).

# 6 Limitations

While our study has made significant strides in its domain, we acknowledge certain limitations that present themselves as opportunities for future research and optimization. Firstly, we have not yet benchmarked our model with other encoder-decoder frameworks such as BART, or with alternative PEFT methods like LORA, leaving room for potential performance enhancement. We believe that the diversity could further improve with diversity aware coherence loss (Li et al., 2023). Secondly, our model has yet to reach the full potential of FlanT5 due to current model size constraints. This implies that scaling up the model could further improve its performance. Thirdly, we have not fine-tuned the number of dimensions for the CNN encoder output or explored structures beyond basic CNN, LSTM, and MLP, both of which could enhance our current performance. Fourthly, We noted a relatively high variance in DeTiME's performance, we interpret this as a consequence of the complicated autoencoder structure. Lastly, we have not benchmarked all coherence metrics. Though many metrics have similarities and some may not consider semantic word meaning, a more extensive benchmarking could provide a richer evaluation of our approach. Despite these limitations, each of these points serves as a promising direction for future research, thereby helping to further elevate our model's capabilities.

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

# Appendix

## A  Qualitative study

In this appendix, we mainly discuss how we set up the qualitative survey for diffusion-based text generation.

As mentioned in the main content, we mainly measure the fluency, grammar, and redundancy of the generated text. Based on this reference (Celikyilmaz et al., 2021), we have designed the corresponding questions in Table. 3. For each question, five answer options are listed from strong negative to strong positive, and a score is assigned to each option. In this survey, we have sampled 400 one-hot topic vectors and generated text following the generating component in fig. 3 for each datasets. We then leverage the Amazon Mechanical Turk to evaluate the quality of each generated sentence. In this process, We have requested three independent reviewers to mitigate the individual bias, and the average score is calculated for each sample. The histogram of the collected scores is shown in fig.4. At the end, we have employed a t-test to evaluate if this survey is statistically significant. The null hypothesis has been tested against the one-sided alternative that the mean of the population is greater than 0 for fluency and grammar, and the null hypothesis against the one-sided alternative that the mean of the population is less than 1 for redundancy. The $p < 1e - 14$ have been obtained and thus we can reject the null hypothesis for all of them.

We use the ratings and word intrusion tasks as human evaluations of topic quality. We recruit crowdworkers using Amazon Mechanical Turk inside Amazon Sagemaker. We pay workers 0.024 per task. We select 3 crowdworkers per task for 400 generated contents per task.

In Table.4 below, we present a comparison between a sample text generated without the denois-

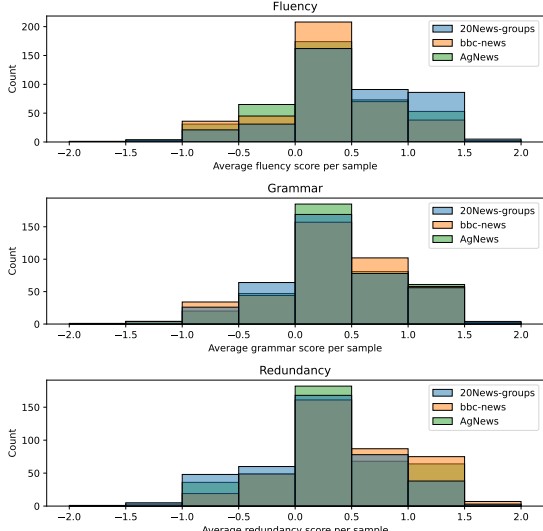

Figure 4: The histogram of survey scores from fluency, grammar, and redundancy perspectives.

ing process and five generated text with denoising from the same topic vector $z''$.

## B  Datasets

We have created a huggingface account to upload all relevant data used for training our modified FlanT5: `https://huggingface.co/datasets/xwjzds/pretrain_sts`

We use the same account to upload all raw data for topic modeling as you can see: `https://huggingface.co/datasets/xwjzds/ag_news`, `https://huggingface.co/datasets/xwjzds/bbc-news`, and `https://huggingface.co/datasets/xwjzds/20_newsgroups`. We have uploaded text after preprocessing here: `https://huggingface.co/datasets/xwjzds/ag_news_lemma_train`, `https://huggingface.co/datasets/xwjzds/bbc-news_lemma_train`, and `https://huggingface.co/datasets/xwjzds/20_newsgroups_lemma_train`. We have uploaded words used for bag of words here: `https://huggingface.co/datasets/xwjzds/ag_news_keywords`, `https://huggingface.co/datasets/xwjzds/bbc-news_keywords`, and `https://huggingface.co/datasets/xwjzds/20_newsgroups_keywords`. Overall, we use 3 datasets that combines different domain to evaluate the performance.

(1) **AgNews** We use the same AG's News dataset from (Zhang et al., 2016).Overall it has 4 classes and, 30000 documents per class. Classes categories

Table 3: The setup of survey

| metrics | question | options |
|---|---|---|
| Fluency | Is the language in the sentence fluent? | • not fluent at all (-2)
• not fluent (-1)
• neutral (0)
• fluent (1)
• very fluent (2) |
| Grammar | How grammatical the generated text is? | • not grammatical at all (-2)
• not grammatical (-1)
• neutral (0)
• grammatical (1)
• perfectly grammatical (2) |
| Redundancy | How repetitive or redundant the generated text is? | • not redundant at all (-2)
• not redundant (-1)
• neutral (0)
• redundant (1)
• very redundant (2) |

Table 4: The generated text with and without denoising

| The text generated without denoising | The text generated with denoising |
|---|---|
| "I'm not sure if this is a good idea or not, but I'm sure it's a good idea." | "the act of removing a bacteriophage from a plant is a source of danger."
"a few years ago a blond man was driving a honda civic car."
"the act of putting a letter or a symbol in a document."
"the man, who is a philanthropist, died in a car crash in april, 2000."
"the act of stealing a car." |

include World, Sports, Business, and Sci/Tech.

(2) **bbc-news** (Lang, 1995) has 2225 texts from bbc news, which consists of 5 categories in total. The five categories we want to identify are Sports, Business, Politics, Tech, and Entertainment.

(3) **20Newsgroups** (Lang, 1995) is a collection of newsgroup posts. We only select 20 categories here. Compare to the previous 2 datasets, 20 categories newsgroup is small so we can check the performance of our methods on small datasets. Also, the number of topics is larger than the previous one.

## C  Code

**Code for our architecture** $Enc_1$ and $Enc_2$ is modified from `https://huggingface.co/transformers/v3.0.2/_modules/transformers/modeling_t5.html#T5Model.forward` by modifying the forward process. **Training process for modified T5** We use google/flan-t5-base as our basic model. We use 20 percent data as the validation set. We train 20 epochs or when validation loss deteriorates consistently for 3 epochs. We set the number of virtual tokens equal to 20. We set the learning rate to 0.01. For the CNN encoder we have 2 1-dimension convolution layers with GELU as the activation function. In channel is 256, 32, and 4. Kernal size is 3 and stride is 1 and padding is 1. We set dropout after the convolution layer with a dropout rate equal to 0.2. We have not systematically finetuned these parameters. We trained our modified FlanT5 for 3 times and choose the lowest validation loss model as our model to run topic modeling. It took less than 20 hours for a single gpu to finetune task.

**Code for comparable methods** Code we used to implement GSM is `https://github.com/YongfeiYan/Neural-Document-Modeling` with topic covariance penalty equals to 1. The code we used to implement ETM is `https://github.com/adjidieng/ETM` ntm.py in zip file is where we rewrite and includes all relevant topic modeling methods.

The code we used to implement CTM and ZTM is `https://github.com/MilaNLProc/contextualized-topic-models` For CTM and ZTM, We set the number of samples for topic predictions equal to 5 and used their default preprocessing methods. The code we used to implement vONT is derived from `https://github.com/YongfeiYan/Neural-Document-Modeling` The

code we used to sentence embeddings vectors is from huggingface: https://huggingface.co/sentence-transformers gsm-vae.py is where we implement our version of topic modeling.

**Code for metric diversity** is implemented using scripts: https://github.com/adjidieng/ETM/blob/master/utils.py line 4. $C_v$ is implemented using gensim.models.coherencemodel where coherence = '$C_v$', **Top-NMI** is implemented using metrics.$normalized_mutual_info_score$ from sklearn. **Top-Purity** is implemented by definitions. **km** based is implemented by the sklearn package kmeans.

## D  Compared Methods Selection

**Sentence Embedding** We choose GTR-T5 and Sentence-T5 because they are the only two embeddings that we are aware of T5 as base models. They also perform well in clustering tasks (Muennighoff et al., 2022). We choose Mpnet because it is commonly used in sentence embeddings and is the second best method in clustering. Benchmarking all these methods shows that our method is superior in sentence embeddings when the number of topics is large.

**Topic Modeling** There are many neural topic modeling methods but no standard benchmarks. For neural topic modeling methods, we choose NVDM because it performs well in (Doan and Hoang, 2021). We choose ETM because it is commonly used and is the first one to leverage word embeddings to topic modeling. We choose vONT (Xu et al., 2023d) because it performs well in clusterability topic modeling metrics. We choose GSM because it also applied softmax after sampling from the gaussian distribution. We think it is a similar comparison.

For contextualized topic modeling, we choose CTM and ZTM because they are the best performing ones with code available. We exclude methods such as Topic2Vec or Berttopic because it is hard to define the number of topics or get the embeddings of documents to calculate clusterability. While many methods are derived from CTM, they either do not have code or are hard to use. For example, (Wu et al., 2023) is hard to process data for bbc news in the same format. In the future, we would like to benchmark methods such as (Costello and Reformat, 2023a) which leverage reinforcement learning, (Wang et al., 2023) which leverage adversairal training . Other methods such as (Han et al.,

2023) have no code. We only include methods that leverage language models to do an apple-to-apple comparison and exclude methods using graph neural network (Zhou et al., 2020) or reinforcement learning (Costello and Reformat, 2023b).

## E  Settings for clusterability evaluations

Compare existing sentence embedding methods with our proposed embedding on standard clusterability metrics such as purity and NMI on Ag-News dataset (Zhang et al., 2016). We compare our methods with GTR-T5 (Ni et al., 2021a), Sentence-T5 (Ni et al., 2021b) and Mpnet (Song et al., 2020). We choose the largest version for all of them. De-TiME training is the embedding finetuned on the same dataset but we use the same input and output instead of rephrasing. We train a 2 layer MLP neural network variational autoencoder without softmax suggested by (Miao et al., 2017). We choose the mean to represent the hidden dimension of the input. We train each embedding 10 times to get the confidence band and average. The consistency of DeTiME's clusterability from 20 to 50 epochs suggests its potential suitability for topic modeling for the large number of topics.

## F  the purpose of CNN encoder

The theoretical advantage of CNN is to extract local features, reduce dimension reduction and be robust to noise(Li et al., 2021). In our cases, it helps to further extract important and local features from LLM encoder output and reduce dimensions.

The purpose of CNN is to compress output from FlanT5 encoder to create embeddings for topic modeling. The output of FlanT5 encoder is 256 (maximum sequence length used in our pre-train model) * 768 (embedding dimension) = 393216. To illustrate our points, we rerun the same clusterability experiment but replacing CNN encoder with MLP. The topic purity drops from 0.614 to 0.396 and NMI drops from 0.31 to 0.05. This shows that CNN encoder helps the framework to achieve high clusterability and suitable for topic modeling. For efficiency, since the input dimension of encoder is 393216 and the output is 3072. MLP will require parameters 1207962624 parameters while CNN only reuqires 49624 parameters. This makes CNN is easy to load and much effcient to train.

This dimension is too high for any NTM to extract information. Thus, we need to compress the obtained embeddings for topic modeling. Here,

using MLP only is hard to build representations that incorporate information across the entire input text sequence as we just concatenate the embeddings of each tokens and then fed into MLP. In the same time, the authors in (Beltagy et al., 2020) have shown that a very lightweight convolution can perform competitively to the best reported self-attention results. This shows that CNN is effective on extracting information from attention layer. Based on this, we thus leverage CNN encoder to build representations that capture information across the text sequence. Also, by reducing the number of output channels, we are able to obtain embeddings with reduced dimensions (4 * 768=3072 in our method), which hugely speed up the training.

## G  Perplexity Evaluation

We have measured the perplexity for our method and other methods on the dataset AgNews, and the results are vNTM: 1479.32, ETM 692.17, NVDM: 1734.28, GSM 684.31 DeTiME: 612.71. This strengthen our conclusion of our work that De-TiME is promising in topic modeling. We use the same set up as (Gupta and Zhang, 2021) for this experiment.

Perplexity is not applied in topic-aware content generation and has not been used in topic modeling lately. We had not reported perplexity as the topic-aware content is generated from the sampled latent topic embeddings, where the ground truth (i.e. text sequence ground truth) is not available. The reason we used sampled latent topic embeddings is that we are mainly focus on how diffusion can improve the quality of topic aware text generation.

## H  Related Work

To the best of our knowledge, we are the first to fine-tune and modify encode-decoder LLM (i.e. Flan-T5) in topic modeling, and the first one to use diffusion in topic aware content generation with topic modeling, and integrate both in one unified framework. We do not find simpler structures to solve topic modeling and topic aware generation using encoder decoder LLM. For topic aware content generation using diffusion, there is no comparable work and we have to establish all baselines ourselves for this. We have list other comparable works below and how our work distinguish from them:

(Cui and Hu, 2021) has leveraged Bert as part of encoder. However, they used NTM that taken bag of words as input and Bert/Graph neural network as encoder. We instead use encoder and decoder LLM and we take embeddings from encoder as input to Neural Topic Modeling. Also, their goal is summarization but our goal is topic aware generation. (Ailem et al., 2019) propose a new decoder where the output summary is generated by conditioning on both the input text and the latent topics of the document. The latent topics, identified by a topic model such as LDA, reveals more global semantic information that can be used to bias the decoder to generate words. In our work, we have leveraged a more promising topic modeling based on encoder-decoder LLM. Also, the diffusion model is used to generate high-quality topic aware text, instead of summarization. (Zhang et al., 2023) proposed to directly generating the desired summary sentence representations with diffusion models and extracting sentences based on sentence representation matching. Even though this work leveraged the distribution of embedded vectors of text for matching, it does not leverage the topic modeling. In comparison, our work have leveraged topic modeling, and also is able to generate high-quality topic-aware text instead of extractive summarization.