# OpenReview forum: "DeTiME: Diffusion-Enhanced Topic Modeling using Encoder-decoder based LLM"
_EMNLP/2023/Conference — EMNLP 2023 Findings_

### Official Review · Reviewer_WMyA · 2023-07-23

**Soundness:** 3

**Excitement:**

2: Mediocre: This paper makes marginal contributions (vs non-contemporaneous work), so I would rather not see it in the conference.

**Missing References:**

Lack of research on hierarchical topic models and topic models integrated with neural networks.

**Paper Topic And Main Contributions:**

This paper proposes a framework named Diffusion-Enhanced Topic Modeling using Encoder-Decoder-based LLMs (DeTiME).
DeTiME is designed to generate topics that exhibit both superior clusterability and enhanced semantic coherence compared to existing methods.
DeTiME leverages the strengths of paraphrase tasks, FlanT5, CNN, and a variational autoencoder structure capable of reconstructing embeddings while simultaneously producing highly coherent, diverse, and clusterable topics.
By exploiting the power of diffusion,
DeTiME also provides the capability to generate content relevant to the identified topics.
This dual functionality allows users to efficiently produce highly clustered topics and related content simultaneously.
DeTiME’s potential extends to generating clustered embeddings as well.

**Questions For The Authors:**

Please see the reject.

**Reasons To Accept:**

* Proposed framework, DeTiME, aims to close the gap that Neural Topic Models (NTMs) have leveraged contextual embeddings from LLMs, neglecting the potential benefits of harnessing the overall structure.

* DeTiME is designed to exploit the capabilities of the encoder-decoder Large Language Model (LLM) and train an adapted encoder-decoder LLM.

* Limitation is reasonable

* Easy to read.

**Reasons To Reject:**

* The structure of the framework is complex
  * While DeTiME consists of CNN, VAE, and T5 (encoder and decoder) and employs a diffusion process,
    topic models with simpler structures have been proposed to tackle the same issues.
  * Weak motivation for why these models were chosen and combined in this way.

* Weak literature review and experimentation
  * The paper appears to be well-cited in the existing literature, but cites few models that attempt the same problem.
  * For the above reasons, the number of benchmark models in the experiment is small and the number of evaluation items is also small.

**Reproducibility:**

4: Could mostly reproduce the results, but there may be some variation because of sample variance or minor variations in their interpretation of the protocol or method.

**Reviewer Confidence:**

5: Positive that my evaluation is correct. I read the paper very carefully and I am very familiar with related work.

---

> ### Author Rebuttal · Authors · 2023-08-28
>
> Thanks the reviewer for their thoughtful feedback! I will address them one by one.
>
> 1. Question: While DeTiME consists of CNN, VAE, and T5 (encoder and decoder) and employs a diffusion process, topic models with simpler structures have been proposed to tackle the same issues. The paper appears to be well-cited in the existing literature, but cites few models that attempt the same problem.
>
> Answer: Thanks for your comments. To the best of our knowledge, we are the first to finetune and modify encode-decoder LLM (i.e. Flan-T5) in topic modeling, and the first one to use diffusion in topic aware content generation with topic modeling, and integrate both in one unified framework. We do not find simpler structures to solve topic modeling and topic aware generation using encoder decoder LLM.  For topic aware content generation using diffusion, there is no comparable work and we have to establish all baselines ourselves for this. We have list other comparable works below and how our work distinguish from them:
>
>     [1] has leveraged Bert as part of encoder. However, they used NTM that taken bag of words as input and Bert/Graph neural network as encoder. We instead use encoder and decoder LLM and we take embeddings from encoder as input to Neural Topic Modeling. Also, their goal is summarization but our goal is topic aware generation.
>     [2] propose a new decoder where the output summary is generated by conditioning on both the input text and the latent topics of the document. The latent topics, identified by a topic model such as LDA, reveals more global semantic information that can be used to bias the decoder to generate words. In our work, we have leveraged a more promising topic modeling based on encoder-decoder LLM. Also, the diffusion model is used to generate high-quality topic aware text, instead of summarization.
>     [3] proposed to directly generating the desired summary sentence representations with diffusion models and extracting sentences based on sentence representation matching. Even though this work leveraged the distribution of embedded vectors of text for matching, it does not leverage the topic modeling. In comparison, our work have leveraged topic modeling, and also is able to generate high-quality topic-aware text instead of extractive summarization.
>
> We will add those comparison in revised version. We have done extensive literature on relevant work. If you find any other relevant work that is not covered by this, please share with us. We would love to citing and showing the difference between our work and other work.
> [1] Cui, P., & Hu, L. (2021). Topic-guided abstractive multi-document summarization. arXiv preprint arXiv:2110.11207.
>
> [2] Ailem, Melissa, Bowen Zhang, and Fei Sha. "Topic augmented generator for abstractive summarization." arXiv preprint arXiv:1908.07026 (2019).
>
> [3] Zhang, Haopeng, Xiao Liu, and Jiawei Zhang. "Diffusum: Generation enhanced extractive summarization with diffusion." arXiv preprint arXiv:2305.01735 (2023).
>
> 2. Question: Weak motivation for why these models were chosen and combined in this way.
>
> Answer: Please see the detailed explanations of each modules and their combination:
>
> FlanT5: We need encoder/decoder model to extract the embeddings and generate relevant contents. At the point when we submitted the paper, FlanT5 did good job on both. Thus, we select FlanT5. Please find details in line 295-300.
>
> CNN: This is used to compress the encoder output which has 393216 dimensions. To illustrate our points, we rerun the same clusterability experiment but replacing CNN encoder to MLP. The topic purity drops from 0.614 to 0.396 and NMI drops from 0.31 to 0.05. This shows that CNN encoder help the framework to achieve high clusterability and suitable for topic modeling. Please find details in line 301-319 and answer 1 for the reviewer Y61q
>
> VAE: Like all other neural topic modeling, this is used to generate topics. We chosen the exact structure because it performs well in both topic modeling relevant metrics compared to other variations see Table 1.
>
> Diffusion: The topic modeling part of DeTIME may induce noise during the text reconstruction, and using Diffusion models is helpful to denoise the reconstructed continuous embedding vectors and thus to improve the quality of topic aware generated text. Please find more details in 426-440 and answer 3 in reviewer Y61q
>
> Combination: Encoder Decoder model can give us a good embedding but the dimension is too high. Thus, we put CNN inside Encoder Decoder model. We need VAE to create Neural Topic Modeling method. NTM performs well when the dimension is not too high. Thus, we put it inside CNN encoder. We use diffusion because our latent dimension is low. Thus, we need diffusion to denoise it and generate high quality output.
>
> 3. Question: For the above reasons, the number of benchmark models in the experiment is small and the number of evaluation items is also small.
>
> Answer: For topic modeling, the number of benchmark models in our work is 6 and the number of evaluation metrics are 6 and evaluation datasets are 3. While we agree that there are many models and metric in topic modeling, we believe we covered most of main stream models and important variations, which strongly supports our conclusion in the paper. We explain the detailed reason why we pick these models in line 1172 - 1199.
>
> For content generation, we benchmark our method at different diffusion time stamps across 3 metrics and 3 datasets as well as qualitative study. To the best of our knowledge, we are the first to leverage diffusion in topic aware content generation with topic modeling. Therefore there is no comparable work and we have to establish all baselines ourselves for content generation by diffusion.
>
>
> 4. Question: Lack of research on hierarchical topic models and topic models integrated with neural networks.
>
> Answer: All 6 selected benchmark models are topic models integrated with neural networks as explained in line 456-481. Thanks for bringing hierarchical topic models up, we agree that extending DeTiME to hierarchical topic modeling is our next step.  We exclude hierarchical topic models this time because DeTiME is a general topic modeling method and we only benchmark general topic modeling methods.

---

### Official Review · Reviewer_q8m2 · 2023-08-02

**Soundness:** 4

**Excitement:**

4: Strong: This paper deepens the understanding of some phenomenon or lowers the barriers to an existing research direction.

**Paper Topic And Main Contributions:**


In this paper, the authors present a study of generating embeddings that are suitable for clustering (and topic modeling) tasks. To do this, the authors propose an encoder-decoder LLM that is able to generate clusterable topics and related topic sentences. The authors validate their experimental results on some standard labeled benchmark datasets, including AgNews, 20News, and bbc-news. Experimental results show that the proposed method is outperforming several identified baselines.

In general, I agree with the authors' motivation that the currently available sentence embeddings are not totally suitable for clustering tasks. It is especially true for some domain-specific applications. The idea for conducting such a study is solid. Regarding the methods, although it is not totally new for each module used in the proposed framework. Nonetheless, the framework as a whole can not be treated as simple add-together of different blocks or incremental work. For the experiments, I am fine with the authors' choices of baselines along with the human evaluation. In sum, I think it is a solid work.

**Reasons To Accept:**


I think the authors have a good observation that sentence embeddings do not lead to good clustering results. The proposed framework to me makes sense. The authors are able to back up their claims using experimental results.

Regarding the experimental sections, I do have some suggestions:

1. The authors could consider some visualization methods or whatever to provide some more insights on why the new embeddings are better compared to previous methods.

2. The authors might want to provide more examples to demonstrate the meaning of the clusters and the associated sentences.

3. The authors also might want to apply this method to more datasets to test the generalization of it.

**Reasons To Reject:**


I do not have major reasons to reject this paper.

**Reproducibility:**

4: Could mostly reproduce the results, but there may be some variation because of sample variance or minor variations in their interpretation of the protocol or method.

**Reviewer Confidence:**

3: Pretty sure, but there's a chance I missed something. Although I have a good feel for this area in general, I did not carefully check the paper's details, e.g., the math, experimental design, or novelty.

**Typos Grammar Style And Presentation Improvements:**


Somehow I think the paper is written a little bit rushed. The Appendix is not polished. Also, I understand the authors use the bold fonts to emphasize their points, but I do not think it is necessary to have so much bolded text throughout the whole paper.

Other minor issues include:

- "fig.3" -> "Figure 3" or "Fig. 3".

- line 520 and line 538, a blank space after the bracket

---

> ### Author Rebuttal · Authors · 2023-08-28
>
> We thank the reviewer for the thoughtful review and spending time on understanding our work.  Yes, finding an embedding method for topic modeling is really hard. That is the exact idea that leads to this work! Yes, without any piece of module, the framework won't work.
>
> We will take the suggestions to heart and add more visualization, provide more examples, reduce bold fonts and apply to more datasets in our final draft. We will polish appendix and fix minor issues.

---

### Official Review · Reviewer_Y61q · 2023-08-04

**Soundness:** 3

**Excitement:**

3: Ambivalent: It has merits (e.g., it reports state-of-the-art results, the idea is nice), but there are key weaknesses (e.g., it describes incremental work), and it can significantly benefit from another round of revision. However, I won't object to accepting it if my co-reviewers champion it.

**Missing References:**

1.	Neural Topic Modeling via Discrete Variational Inference (https://dl.acm.org/doi/abs/10.1145/3570509)

**Paper Topic And Main Contributions:**

For neural topic modeling, this paper proposes DeTiME (Diffusion-Enhanced Topic Modeling using Encoder-Decoder-based LLMs). Specifically, the model is built upon the following modules: Flan-T5 (Encoder-Decoder LLM), CNN (Convolutional Neural Network), VAE (Variational Auto-Encoder), and Diffusion. Experiments on 3 datasets: 20Ng, bbc-news and AgNews, show the effectiveness of the proposed model.

**Questions For The Authors:**

1.	What is the purpose of CNN encoder?

2.	Why are there 2 rows for NVDM in Table2?

3.	What is the motivation of using Diffusion models for Topic Modeling?

**Reasons To Accept:**

1.	DeTiME performs better than the neural baseline models.

**Reasons To Reject:**

1.	The paper is loosely written.

      a.	There is no clear motivation on why diffusion models can be utilized for Topic Modeling except the weak claim in Line212-213.

      b.	I am not sure about the purpose of section 3.1 when “Datasets” and “Tasks” are introduced in section 4.1.

      c.	It is difficult to understand the model description from Sections 3.2-3.3. For instance, dec1 is not mentioned in the text except in the caption of Figure2. My suggestion is that the Figure 2 and description in the text should match. The authors are requested to clearly explain all the different modules thoroughly available in Figure2.

2.	A brief description of Diffusion models is missing.

3.	This paper is missing Perplexity which is one of the important metrics to quantify the quality of content generation.

**Reproducibility:**

3: Could reproduce the results with some difficulty. The settings of parameters are underspecified or subjectively determined; the training/evaluation data are not widely available.

**Reviewer Confidence:**

4: Quite sure. I tried to check the important points carefully. It's unlikely, though conceivable, that I missed something that should affect my ratings.

**Typos Grammar Style And Presentation Improvements:**

Please correct the grammar of following sentence. Line 061-062: However, sentence embeddings are in general not perform well in clustering.

---

> ### Author Rebuttal · Authors · 2023-08-28
>
> Thanks the reviewer for their thoughtful feedback! We will address them one by one.
>
> 1. Question: What is the purpose of CNN encoder?
>
> Answer: The purpose of CNN is to compress output from FlanT5 encoder to create embeddings for topic modeling. The output of FlanT5 encoder is 256 (maximum sequence length used in our pre-train model) * 768 (embedding dimension) = 393216. To illustrate our points, we rerun the same clusterability experiment but replacing CNN encoder with MLP. The topic purity drops from 0.614 to 0.396 and NMI drops from 0.31 to 0.05. This shows that CNN encoder helps the framework to achieve high clusterability and suitable for topic modeling.
>
> This dimension is too high for any NTM to extract information. Thus, we need to compress the obtained embeddings for topic modeling. Here, using MLP only is hard to build representations that incorporate information across the entire input text sequence as we just concatenate the embeddings of each tokens and then fed into MLP. In the same time, the authors in [1, 2] have shown that a very lightweight convolution can perform competitively to the best reported self-attention results. This shows that CNN is effective on extracting information from attention layer. Based on this, we thus leverage CNN  encoder to build representations that capture information across the text sequence. Also, by reducing the number of output channels, we are able to obtain embeddings with reduced dimensions (4 * 768=3072 in our method), which hugely speed up the training. We also explained more in line 301-319
>
>
>
> [1] Beltagy, Iz, Matthew E. Peters, and Arman Cohan. "Longformer: The long-document transformer." arXiv preprint arXiv:2004.05150 (2020).
>
> [2] Wu, Felix, et al. "Pay less attention with lightweight and dynamic convolutions." arXiv preprint arXiv:1901.10430 (2019).
>
>
>
> 2. Question: This paper is missing Perplexity which is one of the important metrics to quantify the quality of content generation.
>
> Answer: We have run additional experiments to show the effectiveness of our methods in perplexity.
>
> We have measured the perplexity for our method and other methods on the dataset AgNews, and the results are vNTM: 1479.32, ETM 692.17, NVDM: 1734.28, GSM 684.31 DeTiME: 612.71. This strengthen our conclusion of our work that DeTiME is promising in topic modeling. We use the same set up as [1, 2] for this experiment.
>
> Perplexity is not applied in topic-aware content generation and has not been used in topic modeling lately. We had not reported perplexity as the topic-aware content is generated from the sampled latent topic embeddings, where the ground truth (i.e. text sequence ground truth) is not available. The reason we used sampled latent topic embeddings is that we are mainly focus on how diffusion can improve the quality of topic aware text generation.
>
> For topic modeling, we agreed that the perplexity is an important metric for topic modeling in LDA [3], but this metric has not been used recently. By search the papers title using key words topic model, we found 0/7 EMNLP 2022 papers have include perplexity in their work. Specifically, in [4], it mentioned that perplexity actually correlated negatively with human determinations of coherence as estimated using behavioral measures. To compare, 5 of them have used topic coherence and/or topic diversity, which have been included in our work.
>
>
> [1] Amulya Gupta and Zhu Zhang. 2021. Vector-Quantization-Based Topic Modeling. ACM Trans. Intell. Syst. Technol. 12, 3, Article 34 (June 2021), 30 pages. https://doi.org/10.1145/3450946
>
> [2] Amulya Gupta and Zhu Zhang. 2023. Neural Topic Modeling via Discrete Variational Inference.  ACM Transactions on Intelligent Systems and Technology 14.2: 1-33.
>
>
> [3] David M. Blei, Andrew Y. Ng, and Michael I. Jordan. 2003. Latent dirichlet allocation. J. Mach. Learn. Res. 3, null (3/1/2003), 993–1022.
>
> [4] Alexander Miserlis Hoyle, Pranav Goel, Rupak Sarkar, and Philip Resnik. 2022. Are Neural Topic Models Broken?. In Findings of the Association for Computational Linguistics: EMNLP 2022, pages 5321–5344, Abu Dhabi, United Arab Emirates. Association for Computational Linguistics.
>
>
>
>
> 3. Question: What is the motivation of using Diffusion models for Topic Modeling?There is no clear motivation on why diffusion models can be utilized for Topic Modeling except the weak claim in Line 212-213.
>
> Answer:  The motivation of using diffusion models is to improve the quality of generated contents. The topic modeling part of DeTIME may induce noise during the text reconstruction, and using Diffusion models is helpful to denoise the reconstructed continuous embedding vectors and thus to improve the quality of topic aware generated text. Specifically, the latent dimension is 20 but the input length of FlanT5 decoder is 393216 in DeTIME. When we reconstruct text from latent vectors, it may hugely deviate from any reasonable input for FlanT5 decoder. Diffusion can thus help through denoising to find the most closed and suitable embeddings for FlanT5 decoder . That is, we are mostly leveraging the diffusion to improve the quality of topic-aware text generation. Several papers[1,2,3,4] have demonstrated the advantages using diffusion model to generate high-quality text from noise samples in the continuous embedding space. We gave more detailed explanations in line 426-440.
>
> [1] Li, Xiang, et al. "Diffusion-lm improves controllable text generation." Advances in Neural Information Processing Systems 35 (2022): 4328-4343.
>
> [2] Gong, Shansan, et al. "Diffuseq: Sequence to sequence text generation with diffusion models." arXiv preprint arXiv:2210.08933 (2022).
>
> [3] Gao, Zhujin, et al. "Difformer: Empowering diffusion model on embedding space for text generation." arXiv preprint arXiv:2212.09412 (2022).
>
> [4] Lin, Zhenghao, et al. "Text Generation with Diffusion Language Models: A Pre-training Approach with Continuous Paragraph Denoise" arXiv preprint arXiv:2212.11685 (2023).
>
> 4. Question: A brief description of Diffusion models is missing.
>
> Answer: Thanks for your comments. We have introduce DDPM in line 199-213. Also, we have documented the detail of diffuser in DeTIME in line 414-424. We agree that we need more details for diffusion and will add more references about using diffusion in text generation and corresponding brief descriptions in the final version.
>
> 5. Question: I am not sure about the purpose of section 3.1 when “Datasets” and “Tasks” are introduced in section 4.1.
>
> Answer: Section 3.1 explains datasets for pre-training. Section 4.1 are datasets for downstream tasks. We recognize how the current structure of the paper makes this distinction confusing.  To address this,  we will restructure the paper to introduce the datasets in section 3.
>
> 6. Question: 1. It is difficult to understand the model description from Sections 3.2-3.3. For instance, dec1 is not mentioned in the text except in the caption of Figure2. My suggestion is that the Figure 2 and description in the text should match. The authors are requested to clearly explain all the different modules thoroughly available in Figure2.
>
> Answer: we agree and will edit figure 2 and description to match in the final version.
>
> 7. Question: Why are there 2 rows for NVDM in Table2?
>
> Answer: it is a typo, the second one should be GSM. We will edit it in the final version.
>
> 8. Question: Neural Topic Modeling via Discrete Variational Inference (https://dl.acm.org/doi/abs/10.1145/3570509)
>
> Answer: Thanks for bring this reference to us. We believe this is an important work and will cite this paper in the final version. We will also benchmark DVITM_{full} in our final draft when the code is available.

---

### Meta-Review · Area_Chair_rXfE · 2023-09-22

**Recommendation:** 3

**Metareview:**

Almost all the reviewers have agreed that this is an interesting paper and has shown some good observations.

However, concerns have been raised, though not that critical, on the lacking details and experimental designs.

During the rebuttal period, the authors have partly addressed these concerns and only left some long-term concerns.

---

### Decision · Program_Chairs · 2023-10-07

**Decision:**

Accept-Findings

**Comment:**

Almost all the reviewers have agreed that this is an interesting paper and has shown some good observations.

However, concerns have been raised, though not that critical, on the lacking details and experimental designs.

During the rebuttal period, the authors have partly addressed these concerns and only left some long-term concerns.